# To treat or not to treat: Assessing the role of anti-enterococcal therapy for intra-abdominal infections in patients with cancer

Nana Akazawa[1,2], Naoya Itoh[2]*, Fumika Mano-Usui[3], Hisato Tatsuoka[3,4], Norihiko Terada[5,6], Hanako Kurai[1]

1 Division of Infectious Diseases, Shizuoka Cancer Center, Sunto-gun, Shizuoka, Japan, 2 Division of Infectious Diseases, Aichi Cancer Center, Nagoya, Japan, 3 General Incorporated Association Kansai Healthcare Science Informatics, Kyoto, Japan, 4 Yutaka Seino Distinguished Center for Diabetes Research, Kansai Electric Power Medical Research Institute, Kobe, Japan, 5 Department of Infectious Diseases, Faculty of Medicine, University of Tsukuba, Tsukuba, Ibaraki, Japan, 6 Division of Infectious Diseases, Department of Medicine, Tsukuba Medical Center Hospital, Tsukuba, Ibaraki, Japan

* itohnaoya0925@ybb.ne.jp

**Data Availability Statement:** All relevant data are within the paper and its Supporting Information files.

## Abstract

The clinical significance of enterococci in intra-abdominal infections, particularly those caused by multiple organisms, remains unclear. There are no definitive guidelines regarding the use of empiric therapy with antimicrobial agents targeting enterococci. In this study, we evaluated the impact of the initial antimicrobial therapy administration of anti-enterococcal agents on the treatment of intra-abdominal infections in patients with cancer in whom enterococci were isolated from ascitic fluid cultures. This retrospective study was conducted at Shizuoka Cancer Center between January 1, 2014, and December 31, 2020, on all adult patients with cancer with enterococci in their ascitic fluid cultures. The primary outcome was all-cause mortality, and the secondary outcomes were composite outcomes consisting of three components (mortality, recurrence, and treatment failure) and the risk factors associated with all-cause mortality and composite outcomes. In total, 103 patients were included: 61 received treatment covering enterococci, and 42 did not. The mortality rates did not differ significantly between the treated and untreated groups (treated: 8/61 [13.1%]; untreated: 5/42 [11.9%]; p = 1.00). Additionally, no significant difference was observed between the groups in terms of composite outcomes (treated group: 11/61 [18.0%]; untreated group: 9/42 [21.4%]; p = 0.80). Multivariate analysis showed that performance status (PS2–4; p < 0.0001) was an independent risk factor for mortality. The composite outcome was also significantly higher for PS2–4 (p = 0.007). Anti-enterococcal treatment was not associated with mortality or the composite outcome. In patients with cancer and intra-abdominal infections caused by enterococci, anti-enterococcal therapy was not associated with prognosis, whereas PS2 or higher was associated with prognosis. The results of this study suggest that the initial routine administration of anti-enterococcal agents for intra-abdominal infections may not be essential for all patients with cancer. To substantiate these findings, validation by a prospective randomized trial is warranted.

**Funding:** The author(s) received no specific funding for this work.

**Competing interests:** The authors have declared that no competing interests exist.

## Introduction

Enterococci are gram-positive, catalase-negative, facultative anaerobes that inhabit the intestinal tracts of humans and animals [1]. Enterococci rarely cause complications in healthy individuals because they are not inherently highly pathogenic [1]. However, they are important causes of hospital-acquired infections, such as urinary tract infections, surgical wound infections, bacteremia, and infections that cause cholangitis, endocarditis, and peritonitis [2].

Enterococci are found in 20%–30% of ascitic fluid cultures from patients with intra-abdominal infections [3]; they are the most common gram-positive cocci in nosocomial infections [4]. Moreover, intra-abdominal infections, in which enterococci are isolated from ascitic fluid, have a poor prognosis and high mortality rates [5, 6]. However, it is unclear whether enterococcal infections worsen patient prognosis or whether patients with poor prognosis are more likely to develop enterococcal infections. The clinical significance of enterococci in intra-abdominal infections, particularly those caused by multiple organisms, remains unclear. Consequently, there are no definitive guidelines regarding which patients should receive empiric therapy with antimicrobial agents targeting enterococci before culture results are known. Additionally, there is no consensus on whether treatment with anti-enterococcal drugs can improve patient outcomes [5, 7–10]. Theunissen et al. found that the presence of enterococci was independently associated with increased mortality in both community-acquired and hospital-acquired intra-abdominal infections in immunocompromised patients [6]. Guidelines from the Surgical Infectious Diseases Society and the Infectious Diseases Society of America recommend the use of anti-enterococcal agents as the first-line treatment for intra-abdominal infections in immunocompromised patients [11]. However, these reports only highlight that immunosuppressed patients undergoing transplantation, cancer treatments, or medical treatments for inflammatory diseases are at an increased risk of complications. They stop short of providing a detailed definition of "immunodeficiency."

To the best of our knowledge, no studies have specifically examined intra-abdominal infections caused by enterococci in patients with cancer. Therefore, this study aimed to determine whether the initial administration of agents effective against enterococci before culture results are known is crucial for the treatment of intra-abdominal infections in immunocompromised patients with cancer. Accordingly, we investigated whether there is a difference in mortality, recurrence rate, and treatment failure with or without treatment. Moreover, we examined the prognostic predictors of intra-abdominal infections in patients with cancer in whom enterococci were detected.

## Materials and methods

### Study design, setting and patient population

The Shizuoka Cancer Center is a 615-bed tertiary care hospital. On average, the surgical ward admits approximately 8,000 patients annually, whereas the internal medicine ward admits approximately 7,000. We conducted a single-center retrospective observational study at the Shizuoka Cancer Center. Using data from the Microbiology Department, we reviewed the medical records of all adult patients with cancer with positive intra-abdominal cultures for enterococci between January 1, 2014, and December 31, 2020.

### Inclusion criteria

The inclusion criteria were patients with cancer who developed secondary peritonitis (including cholecystitis, appendicitis, lymphocyst infection, and deep surgical site infection) or an intra-abdominal abscess and had enterococci detected in their puncture (sonographic puncture) or intraoperative ascitic fluid cultures.

## Exclusion criteria

The exclusion criteria were age <18 years, intra-abdominal infection without microbiological culture, no intra-abdominal infection (e.g., routine drain culture after surgery), patient on antimicrobials at the time of culture collection, absence of aggressive treatment for intra-abdominal infection, and unknown outcomes. In addition, we excluded cases involving the use of meropenem and levofloxacin for the following reasons: 1) The Clinical and Laboratory Standards Institute (CLSI) specifies that levofloxacin should only be used for the treatment of urinary tract infections when enterococci are involved, and 2) Meropenem was excluded because it does not fall under group A/B of the drugs recommended for susceptibility testing according to the CLSI guidelines, making it difficult to determine the appropriateness of meropenem treatment [12]. No patients in this study received levofloxacin.

## Study groups

Patients were defined as "receiving effective initial therapy for enterococci" if antimicrobial therapy was initiated within 48 hours of ascitic fluid collection, the enterococci detected in the ascitic fluid were susceptible to that antimicrobial therapy, and treatment was continued for at least 4 days. Patients were divided into two groups, with the group that received the above-defined effective initial therapy against enterococci classified as "treated patients" and the group that did not receive treatment defined as "untreated patients". In cases where multiple types of enterococci were detected in the ascitic fluid, if one or more types of enterococci were present that were not covered by appropriate treatment, the patient was classified as part of the "untreated patients" group.

## Data collection and definitions

Medical records were collected for up to 30 days after completion of treatment for intra-abdominal infection or up to discharge from the hospital. This information was entered into an electronic case report form. The collected data included the following: age, sex, underlying disease (tumor type), cancer stage, performance status (PS), name of the infectious disease, clinical symptoms, presence of bacteremia, surgeries performed within 30 days, the severity of intra-abdominal infection (as measured by the Pitt bacteremia score), microbiological characteristics of ascites fluid culture (species of enterococci, other detected bacteria), treatment method (type of antimicrobial agent, susceptibility to antimicrobial therapy against enterococci species), mortality, recurrence rate, and treatment failure rate. Intra-abdominal infections were defined as secondary peritonitis or abscesses. Secondary peritonitis was defined as peritonitis due to bacterial leakage into the abdominal cavity, primarily due to perforation of the intestinal tract, requiring surgery or percutaneous drainage. Secondary peritonitis also included cholecystitis, appendicitis, lymphocytic infection, and deep surgical site infections such as anastomotic leakage. The tumors were classified as solid or hematological. PS was classified on a scale of 0–4 as defined by the Eastern Cooperative Oncology Group [13]. The severity of intra-abdominal infection was graded from 0 to 14 using the Pitt bacteremia score [14]. Mortality was defined as death from all causes within 30 days of diagnosis. If the treatment duration was >30 days, mortality was defined as death from all causes within 30 days of treatment completion. Recurrence was defined as the development of a new intra-abdominal infection within 30 days of completing treatment for an initial intra-abdominal infection, including cases in which enterococci were not detected in the ascites fluid. Treatment failure was defined as a lack of clinical improvement after completing appropriate treatment for enterococci, as determined by an infectious disease physician. We accessed these data on January 13, 2023, for research purposes.

## Microbiological methods

Ascitic fluid samples were stained using Gram stain and cultured on HK semi-fluid plates (Kyokuto Pharmaceutical Co., Tokyo, Japan), Brucella HK (RS) agar plates (Kyokuto Pharmaceutical Co.), chocolate agar plates No. 2 (Kyokuto Pharmaceutical Co.), and Trisoy blood agar plates (sheep) No. 2 (Kyokuto Pharmaceutical Co.). Depending on the results of Gram staining, a DHL (Deoxycholate Hydrogen sulfide Lactose) agar plate (Kyokuto Pharmaceutical Co., Tokyo, Japan) and sheep blood plate (Nissui Pharmaceutical Co., Ltd., Tokyo) were added to the culture. Enterococci cultured on blood agar plates were identified by the Micro-Scan WalkAway 40 plus System (Beckman Coulter Japan, Tokyo, Japan) until November 6, 2016, and, thereafter, matrix-assisted laser desorption ionization-time-of-flight mass spectrometry on a MALDI Biotyper (Bruker Daltonics Co. Ltd., Billerica, MA, USA). Susceptibility testing was performed using the MicroScan WalkAway 40 plus System until October 14, 2018; thereafter, the DxM 1096 Microscan WalkAway (Beckman Coulter Inc., Carlsbad, CA, USA). The interpretive criteria used to determine antibiotic susceptibility were in accordance with the CLSI's susceptibility testing standards for enterococci, document M100-S22 (2012) [15]. M100-S22(2012) has not changed breakpoints with ampicillin and vancomycin by the current M100-Ed33(2023) [16].

## Outcomes

The primary outcome was all-cause mortality, and the secondary outcomes were composite outcomes consisting of three components (mortality, recurrence, and treatment failure) and the risk factors associated with all-cause mortality and composite outcomes.

## Statistical analysis

Categorical variables were analyzed using Pearson's chi-square test or Fisher's exact test, and Student's t-test was performed for continuous variables. Statistical significance was set at $p < 0.05$. The risk factors for the primary and composite outcomes were analyzed using logistic regression analysis. Multivariate logistic regression analysis included PS, Pitt bacteremia score, and anti-enterococcal therapy, which are important variables that are clinically implicated in outcomes. All statistical analyses were performed using JMP version 17 (SAS Institute, Cary, NC, USA).

## Ethical approval and consent to participate

This study was approved by the Institutional Review Board of the Shizuoka Cancer Center (approval number: J2020-139-2020-1-3) and conducted according to the principles of the Declaration of Helsinki. The requirement for informed consent was waived because this study only used data from electronic medical records.

## Results

### Patient characteristics

Between 2014 and 2020, 103 patients were diagnosed with intra-abdominal infections in which enterococci were detected in ascitic fluid cultures (Fig 1). Of these, 61 patients received treatment covering enterococci, whereas 42 did not. Of the 42 patients, 37 had no antimicrobials covering enterococci, and 5 had antimicrobials covering enterococci for less than 4 days. The characteristics of the patients in both groups are presented in Table 1. The mean age was 66.1 years in the treated group and 69.9 years in the untreated group; however, this difference was not statistically significant (p = 0.097). Overall, only three patients had hematologic

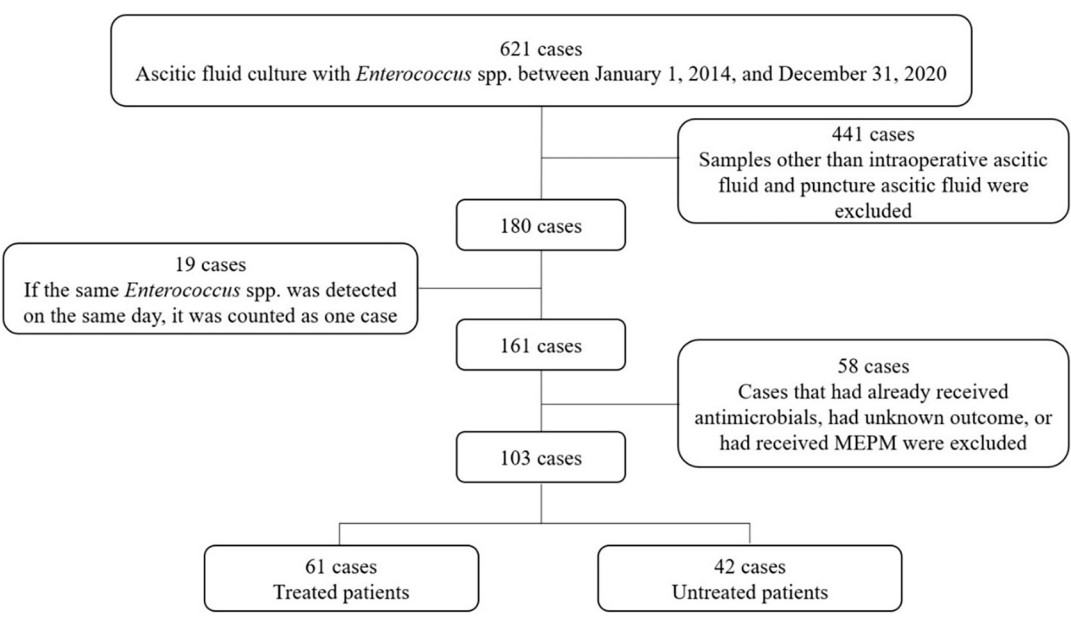

**Fig 1. Flowchart of the patient selection process.** MEPM: Meropenem.

tumors (n = 3, 2.9%), and the majority had solid tumors (n = 100, 97.1%). More than half of the patients in both groups were at stage 3 or higher, and no significant differences were observed between the groups in this regard (p = 0.22). Similarly, no significant difference between the groups was found regarding PS (p = 0.41). Most patients had Pitt bacteremia scores ranging from 0 to 3. Although bacteremia was more common in the treated group, the between-group difference was not statistically significant (treated group: 11/61 [18.0%]; untreated group: 2/42 [4.8%]; p = 0.07). The two groups differed in the type of enterococci present, with a higher percentage of *Enterococcus faecalis* in the treated group and a higher percentage of *E. faecium* in the untreated group (p = 0.0007). The incidence of multiple isolated microorganisms was significantly higher in the treated group than in the untreated group (59/61 [96.7%] vs. 33/42 [78.6%]; p < 0.01). The mortality rates did not differ significantly between the treated and untreated groups (8/61 [13.1%] vs 5/42 [11.9%]; p = 1.00). Additionally, no significant difference was observed between the groups in terms of composite outcomes (treated group: 11/61 [18.0%]; untreated group: 9/42 [21.4%]; p = 0.80).

## Risk factors affecting outcomes of intra-abdominal infections in which enterococci were detected

Univariate analysis showed that female sex (odds ratio [OR] 4.28, 95% confidence interval [CI] 1.22–15.0; p = 0.02), hematologic tumor (OR 16.2, 95% CI 1.35–193.4; p = 0.03), PS2–4 (OR 19.0, 95% CI 4.88–73.8; p < 0.0001), and no previous surgery within the past 30 days (OR 0.05, 95% CI 0.006–0.37; p = 0.004) were risk factors for mortality (Table 2). Multivariate analysis showed that PS2–4 (OR 5.09, 95% CI 2.11–12.3; p < 0.0001) was an independent risk factor for mortality (Table 2).

Univariate analysis showed that PS2–4 (OR 7.24, 95% CI 2.22–23.63; p = 0.001) and no previous surgery within the past 30 days (OR 0.32, 95% CI 0.15–0.89; p = 0.03) were risk factors for the composite outcome, whereas multivariate analysis showed that PS2–4 was a risk factor (OR 2.14, 95% CI 1.20–3.80; p = 0.007; Table 3). Anti-enterococcal treatment was not associated with mortality or the composite outcomes.

**Table 1. Comparison of the clinical characteristics of the patients with and without treatment covering enterococci.**

| | Treated patients n = 61 | Untreated patients n = 42 | p-value |
|---|---|---|---|
| Age (years), mean (SD) | 66.1 (11.4) | 69.9 (10.9) | 0.097 |
| Sex, female | 23 (37.7%) | 17 (40.5%) | 0.84 |
| Underlying cancer | | | |
| Hematologic malignancies | 1 (1.6%) | 2 (4.8%) | 0.57 |
| Solid tumors | 60 (98.4%) | 40 (95.2%) | |
| Cancer stage | | | |
| 0 | 0 | 3 (7.1%) | 0.22 |
| I | 11 (18.0%) | 11 (26.2%) | |
| II | 11 (18.0%) | 6 (14.3%) | |
| III | 22 (36.1%) | 11 (26.2%) | |
| IV | 17 (27.9%) | 11 (26.2%) | |
| Performance status | | | |
| 0 | 23 (37.7%) | 9 (21.4%) | 0.41 |
| 1 | 30 (49.2%) | 26 (61.9%) | |
| 2 | 5 (8.2%) | 4 (9.5%) | |
| 3 | 2 (3.3%) | 1 (2.4%) | |
| 4 | 1 (1.6%) | 2 (4.8%) | |
| Bacteremia | 11 (18.0%) | 2 (4.8%) | 0.07 |
| Enterococcal bacteremia | 2 (3.3%) | 0 | 0.51 |
| Surgery within 30 days | 32 (52.5%) | 27 (64.3%) | 0.31 |
| Pitt bacteremia score | | | |
| 0 | 37 (60.7%) | 24 (57.1%) | 0.84 |
| 1 | 8 (13.1%) | 5 (11.9%) | |
| 2 | 14 (23.0%) | 12 (28.6%) | |
| 3 | 1 (1.6%) | 0 | |
| 4 | 1 (1.6%) | 0 | |
| 5 | 0 | 0 | |
| 6 | 0 | 1 (2.4%) | |
| Type of enterococci | | | |
| *Enterococcus faecalis* | 37 (60.7%) | 10 (23.8%) | 0.0007 |
| *Enterococcus faecium* | 8 (13.1%) | 14 (33.3%) | |
| Other* | 16 (26.2%) | 18 (42.9%) | |
| Isolated multiple microorganisms | 59 (96.7%) | 33 (78.6%) | 0.0067 |
| Mortality | 8 (13.1%) | 5 (11.9%) | 1.00 |
| Recurrence | 2 (3.3%) | 1 (2.4%) | 1.00 |
| Treatment failure | 3 (4.9%) | 5 (11.9%) | 0.27 |
| Composite outcome | 11 (18.0%) | 9 (21.4%) | 0.80 |

Abbreviations: SD, standard deviation

*: *Enterococcus avium*, *Enterococcus casseliflavus*, *Enterococcus gallinarum*, *Enterococcus raffinosus*, *Enterococcus* sp.

## Discussion

To the best of our knowledge, this is the first report to assess the effect of anti-enterococcal therapy in patients with cancer with intra-abdominal infections in which enterococci were isolated from ascitic fluid cultures. We found that anti-enterococcal therapy was not associated with prognosis in cancer patients with intra-abdominal infections caused by enterococci, whereas PS $\geq 2$ was associated with prognosis.

**Table 2. Risk factors for mortality in patients with cancer and intra-abdominal infections with enterococci.**

| | Univariate analysis | | | Multivariate analysis | | |
|---|---|---|---|---|---|---|
| | OR | 95% CI | p-value | OR | 95% CI | p-value |
| Age ≥ 65 years | 0.96 | 0.27–3.40 | 0.95 | - | - | - |
| Female | 4.28 | 1.22–15.0 | 0.02 | - | - | - |
| Underlying cancer | | | | - | - | - |
| Hematologic malignancies | 16.2 | 1.35–193.4 | | | | |
| Solid tumors | 0.06 | 0.005–0.74 | 0.03 | | | |
| Cancer Stage | | | | - | - | - |
| 0–II | 0.61 | 0.17–2.12 | | | | |
| III–IV | 1.64 | 0.47–5.74 | 0.44 | | | |
| Performance status | | | | 5.09 | 2.11–12.3 | <0.0001 |
| 0–1 | 0.05 | 0.01–0.20 | | | | |
| 2–4 | 19.0 | 4.88–73.8 | <0.0001 | | | |
| Bacteremia | 1.31 | 0.25–6.69 | 0.75 | - | - | - |
| Surgery within the past 30 days | 0.05 | 0.006–0.37 | 0.004 | - | - | - |
| Pitt bacteremia score | | | | 1.23 | 0.61–2.49 | 0.56 |
| 0–1 | 0.58 | 0.17–1.95 | | | | |
| 2–6 | 1.72 | 0.51–5.77 | 0.38 | | | |
| Type of enterococci | | | | - | - | - |
| *Enterococcus faecalis* | 0.68** | 0.20–2.34 | 0.54 | | | |
| *Enterococcus faecium* | 0.22*** | 0.02–1.99 | 0.18 | | | |
| Other* | | | | | | |
| Anti-enterococcal treatment | 1.12 | 0.34–3.68 | 0.86 | 1.93 | 0.42–9.02 | 0.40 |

*: *Enterococcus avium*, *Enterococcus casseliflavus*, *Enterococcus gallinarum*, *Enterococcus raffinosus*, *Enterococcus* sp.

**: Odds ratio of *E. faecalis* to other

***: Odds ratio of *E. faecium* to other

Enterococcal bacteremia could not be analyzed because none of the patients who had the primary outcome suffered from enterococcal bacteremia. It was impossible to analyze the multiple microorganisms isolated because all patients who reached the primary outcome had multiple microorganisms detected in their ascitic fluid.

OR, odds ratio; CI, confidence interval

In this study, anti-enterococcal therapy was not associated with mortality or composite outcomes. Although there are limited studies on the relationship between empiric anti-enterococcal therapy and prognosis, some findings are noteworthy. The only prospective randomized controlled trial concluded that prognosis was not affected by the presence or absence of appropriate therapy for enterococci [10]. Conversely, a study focusing on critically ill patients admitted to the intensive care unit with severe peritonitis, in which enterococci were detected, showed that a lack of appropriate treatment was associated with increased 30-day mortality [9]. Additionally, given the high likelihood of detecting enterococci in postoperative intra-abdominal infections, studies have reported higher mortality rates when appropriate anti-enterococcal medications were not administered [5]. A meta-analysis including 23 randomized controlled trials and 13 observational studies [17] showed no improvement in mortality with empiric use of anti-enterococcal agents. Most of the studies included in the meta-analysis were conducted in patients with non-severe community-onset intra-abdominal infections. Although there was no difference in mortality with empiric therapy for patients with cancer, malignancies were associated with a higher risk of enterococcal infections. Therefore, based on the results of these studies, we propose the empiric administration of anti-enterococcal agents from the beginning only for intra-abdominal infections in severely ill patients with cancer.

**Table 3. Risk Factors for composite outcomes.**

| | Univariate analysis | | | Multivariate analysis | | |
|---|---|---|---|---|---|---|
| | OR | 95% CI | p-value | OR | 95% CI | p-value |
| **Age ≥ 65 years** | 0.58 | 0.21–1.59 | 0.29 | - | - | - |
| **Female** | 2.28 | 0.85–6.12 | 0.10 | - | - | - |
| **Underlying cancer** | | | | - | - | - |
| Hematologic malignancies | 9.11 | 0.78–106.0 | | | | |
| Solid tumors | 0.11 | 0.009–1.28 | 0.08 | | | |
| **Cancer Stage** | | | | - | - | - |
| 0–II | 0.74 | 0.27–2.04 | | | | |
| III–IV | 1.35 | 0.49–3.74 | 0.56 | | | |
| **Performance status** | | | | 2.14 | 1.20–3.80 | 0.007 |
| 0–1 | 0.14 | 0.04–0.45 | | | | |
| 2–4 | 7.24 | 2.22–23.63 | 0.001 | | | |
| **Bacteremia** | 0.73 | 0.15–3.58 | 0.70 | - | - | - |
| **Surgery within 30 days** | 0.32 | 0.12–0.89 | 0.03 | - | - | - |
| **Pitt bacteremia score** | | | | 1.26 | 0.77–2.05 | 0.36 |
| 0–1 | 0.67 | 0.24–1.90 | | | | |
| 2–6 | 1.49 | 0.53–4.22 | 0.45 | | | |
| **Type of enterococci** | | | | - | - | - |
| *Enterococcus faecalis* | 1.04 | 0.35–3.09 | 0.94 | | | |
| *Enterococcus faecium* | 0.61 | 0.14–2.66 | 0.51 | | | |
| **Other*** | | | | | | |
| **Isolated multiple microorganisms** | 2.60 | 0.31–21.6 | 0.38 | - | - | - |
| **Anti-enterococcal treatment** | 0.81 | 0.30–2.16 | 0.67 | 0.99 | 0.34–2.85 | 0.98 |

*:*Enterococcus avium, Enterococcus casseliflavus, Enterococcus gallinarum, Enterococcus raffinosus, Enterococcus* sp.

Enterococcal bacteremia could not be analyzed because none of the patients who had composite outcomes suffered from enterococcal bacteremia.

OR, odds ratio; CI, confidence interval

This study showed that PS was associated with both mortality and composite outcomes. This study did not perform a subgroup analysis of PS values and outcomes due to the small sample size. PS is often used as a predictive measure for treatment decisions and prognosis in patients with cancer, as it serves as an important indicator of overall health status and the ability to perform daily activities. It can also predict important clinical outcomes such as quality of life, chemotherapy tolerance, and survival [18]. PS has been identified as an important prognostic factor in patients with various types of cancers [18, 19]. In patients with solid tumors, some reports suggest that PS is a more significant prognostic factor than Sequential Organ Failure Assessment, quick Sequential Organ Failure Assessment, or Systemic Inflammatory Response Syndrome [20]. A study focusing on patients with solid tumors and suspected infections found that cancer progression was not a prognostic factor, whereas PS was, thereby corroborating the findings of our study [20]. These results suggest that PS should be considered when deciding the administration of anti-enterococcal therapy in patients with cancer.

The 30-day mortality rate in our study cohort was 12.6%, which is lower than that reported in previous studies [9, 21]. Kaffarnik et al. reported a 30-day mortality rate of 29.3% for intra-abdominal infections (not limited to enterococci) in immunocompromised patients [21]. Furthermore, Morvan et al. found that the 30-day mortality rate of intra-abdominal infections caused by enterococci was approximately 20% and was higher in immunocompromised patients [9]. This discrepancy may be attributed to the lower severity of illness in our cohort,

as indicated by most patients having a Pitt bacteremia score of ≤2, compared with higher severity scores in previous reports. Additionally, a study focusing on patients with high-risk peritonitis showed that disease severity, rather than the presence of cancer, was associated with mortality [6]. This suggests that cancer alone may not be a significant factor in determining mortality rates.

This study has several limitations. First, this study was performed at a single cancer center, making it unclear whether the results are generalizable. Second, our sample size was relatively small, particularly for patients with hematologic malignancies. Thus, there were limitations in assessing differences in outcomes between solid tumors and hematologic malignancies, as well as differences in outcomes in various cancer types. Concerning patients with hematologic malignancies, for which the sample size was small, the results of this study are not well grounded to be generalized as is. Third, most patients in our cohort had mild abdominal infections. Although previous studies identified disease severity as a significant risk factor for mortality, our study was not adequately powered to assess this because of its small sample size. Also, our study excluded meropenem and levofloxacin, which have negligible activity against enterococci. However, only three cases were excluded, and it is unlikely they could have influenced the study results. Finally, we were unable to evaluate the quality of source control measures, an important factor that affects mortality rates [22]. However, we assumed that infectious disease specialists intervened in most cases and recommended appropriate drainage procedures.

## Conclusion

In patients with cancer who have intra-abdominal infections with enterococci detected in ascites fluid culture, anti-enterococcal therapy was not associated with prognosis, whereas PS2 or higher was associated with prognosis. Our findings suggest that not all cancer patients require initial antimicrobial therapy against enterococci and that treatment should be considered for severe cases and patients with PS2 or higher. Future prospective randomized trials are warranted to confirm these results.

## Supporting information

**S1 File.**
(XLSX)

## Acknowledgments

We thank the staff of the Bacteriology Laboratory at Shizuoka Cancer Center Hospital.

## Author Contributions

**Conceptualization:** Nana Akazawa, Naoya Itoh.

**Data curation:** Nana Akazawa, Norihiko Terada.

**Formal analysis:** Fumika Mano-Usui.

**Investigation:** Nana Akazawa, Naoya Itoh.

**Methodology:** Nana Akazawa, Naoya Itoh, Fumika Mano-Usui, Hisato Tatsuoka.

**Software:** Fumika Mano-Usui.

**Supervision:** Naoya Itoh, Hanako Kurai.

**Writing – original draft:** Nana Akazawa, Naoya Itoh.

**Writing – review & editing:** Nana Akazawa, Naoya Itoh, Fumika Mano-Usui, Hisato Tatsuoka, Norihiko Terada, Hanako Kurai.

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
