## [Decision Letter · Decision Letter 0]

23 Nov 2023

PONE-D-23-31414To treat or not to treat: Assessing the role of anti-enterococcal therapy for intra-abdominal infections in patients with cancerPLOS ONE

Dear Dr. Itoh,

Thank you for submitting your manuscript to PLOS ONE. After careful consideration, we feel that it has merit but does not fully meet PLOS ONE’s publication criteria as it currently stands. Therefore, we invite you to submit a revised version of the manuscript that addresses the points raised during the review process.

We look forward to receiving your revised manuscript.

Kind regards,

Arghya Das, MD

Academic Editor

PLOS ONE

Journal Requirements:

Additional Editor Comments:

Use the term 'ascitic fluid' consistently throughout the manuscript.

Line 62: Ascetic fluid- Please correct the typological error.

Please justify the objective which mentions "empirical administration of agents effective against enterococci". It is important because antimicrobial susceptibility testing was carried out and targeted treatment was given against enterococci at least in one group of patients i.e., "appropriate treatment group". Thereby the anti-enterococcal treatment given to patients cannot exactly be an empiric treatment.

Table 1: The annotations mentioned in row 1 for Age is not clear. What do the statistical figures 66.1 and 69.9 refer to? What statistical method has been adopted to reach to the p value for comparison of age in treated versus untreated groups? Please clarify.

It will be better to mention the groups as "patients with appropriate anti-enterococcal treatment" and "patients without appropriate anti-enterococcal treatment" rather than "Treated patients" and "Untreated patients".

As per Figure 1, if the same Enterococcus species was detected on the same day, it was counted as one case. Does it mean that if same Enterococcus species was detected in same patient on subsequent days, that patient was considered as more than one case? If latter is the true, authors need to relook into the entire statistical calculation. Please clarify.

As per Materials and methods, patients receiving levofloxacin and meropenem were excluded from the study. However, the figure 1 mentions only meropenem treatment as one of the criteria for exclusion.

Reviewers' comments:

Reviewer's Responses to Questions

**Comments to the Author**

1. Is the manuscript technically sound, and do the data support the conclusions?

Reviewer #1: Yes

Reviewer #2: Partly

2. Has the statistical analysis been performed appropriately and rigorously? 

Reviewer #1: Yes

Reviewer #2: Yes

3. Have the authors made all data underlying the findings in their manuscript fully available?

Reviewer #1: Yes

Reviewer #2: Yes

4. Is the manuscript presented in an intelligible fashion and written in standard English?

Reviewer #1: No

Reviewer #2: Yes

5. Review Comments to the Author

Reviewer #1: • Authors have considered a good topic for discussion. However, a sufficient amount of scientific evidence is available on the given research question including a recent systematic review and meta-analysis by Zhang et al. (as mentioned below) with similar conclusions as that of the present study. Authors are requested to review it.

Zhang J, Yu WQ, Chen W, Wei T, Wang CW, Zhang JY, Zhang Y, Liang TB. Systematic Review and Meta-Analysis of the Efficacy of Appropriate Empiric Anti-Enterococcal Therapy for Intra-Abdominal Infection. Surg Infect (Larchmt). 2021 Mar;22(2):131-143. doi: 10.1089/sur.2020.001. Epub 2020 Jun 1. PMID: 32471332.

• The objective of the study as mentioned by the authors in the abstract is “to evaluate the impact of empiric administration of anti-enterococcal agents on the treatment of intra-abdominal infections…” However, as per the study methodology it is evident that, the study measures the impact of definitive therapy guided by culture and sensitivity report showing growth of Enterococcus species and not the empiric therapy. Hence, authors are requested to clarify it and modify the write up at relevant places in the document.

• It is not clear, whether the patients in untreated group received no treatment at all or were given treatment which was ‘inappropriate’ or were given treatment for less than four days.

• Table 2 and 3 - Cancer stage – range parameter not visible

• Kindly give full form for DHL agar plate.

• Since the data collection period is older, interpretative criteria used for antimicrobial susceptibility testing is quite older. A lot of revision has happened in recent years. Authors should take that into account.

• A subgroup analysis could have been done for patients treated with appropriate antimicrobial therapy and having PS-2-4 to substantiate the claim.

Reviewer #2: Dear authors,

I would like to congratulate you for addressing a relevant question in oncology practice . I have added minor suggestions in the document attached.

Lookin forward to your reply.

Best wishes.

6. PLOS authors have the option to publish the peer review history of their article (what does this mean?). If published, this will include your full peer review and any attached files.

Reviewer #1: No

Reviewer #2: **Yes: **Babita Kataria

---

## [Author Response · Author response to Decision Letter 0]

5 Dec 2023

Our point-by-point responses to the comments and suggestions by the Editor are listed below.

Dear Editor:

Thank you for your constructive comments. We have carefully revised our manuscript as suggested, and your insightful comments have helped us significantly improve the quality of our manuscript. The page and line numbers in the responses refer to the relevant parts of the main manuscript (changes marked in red font) where the text has been revised.

Editor:

Comment 1:

Use the term 'ascitic fluid' consistently throughout the manuscript. 

Response 1: Thank you for your valuable suggestions. We have corrected this.

Page 2, line 32：

ascitic fluid cultures.

Page 4, line 60:

Enterococci are found in 20%–30% of ascitic fluid cultures

Page 6, line 104：

intraoperative ascitic fluid cultures.

Page 10, line 161：

Ascitic fluid samples were stained using Gram stain and cultured

Page 12, line 203：

in which enterococci were detected in ascitic fluid cultures (Fig 1).

Figure1

Comment 2:

Line 62: Ascetic fluid- Please correct the typological error.

Response 2: Thank you for pointing this out. We have corrected this.

Page 4, lines 62–63：

Moreover, intra-abdominal infections, in which enterococci are isolated from ascitic fluid,

Comment 3:

Please justify the objective which mentions "empirical administration of agents effective against enterococci". It is important because antimicrobial susceptibility testing was carried out and targeted treatment was given against enterococci at least in one group of patients i.e., "appropriate treatment group". Thereby the anti-enterococcal treatment given to patients cannot exactly be an empiric treatment.

Response 3: Thank you for your very constructive remarks. By "effective empiric therapy for enterococci," I meant "treatment including enterococci as a target was initiated before the ascitic fluid culture was identified, and the patient was treated with that antimicrobial for at least 4 days." We have added clarification where appropriate.

Page 4, lines 67–69：

Consequently, there are no definitive guidelines regarding which patients should receive empiric therapy with antimicrobial agents targeting enterococci before culture results are known.

Page 5, lines 82–85：

Therefore, this study aimed to determine whether the empirical administration of agents effective against enterococci before culture results are known is crucial for the treatment of intra-abdominal infections in immunocompromised patients with cancer.

Page 7, lines 120–123：

"The initiation of therapy that includes enterococci as a target before ascites cultures are known and continues for at least 4 days" was defined as "effective empiric therapy for enterococci." Therefore, we first divided the patients into two groups: those who received treatment covering enterococci and those who did not.

Page 7-8, lines 126–128：

Patients who did not receive treatment covering enterococci were defined as those who did not meet the above definition.

Page 8, lines 129–130：

if one type of enterococci was present and the treatment did not cover that enterococci, the patient was classified in the "untreated patients" group.

Comment 4:

Table 1: The annotations mentioned in row 1 for Age is not clear. What do the statistical figures 66.1 and 69.9 refer to? What statistical method has been adopted to reach to the p value for comparison of age in treated versus untreated groups? Please clarify.

Response 4: Thank you for your valuable suggestions. 66.1 and 69.9 are means. P-values were obtained using Student's t-test. The notation was not clear, so it was changed. The value of standard deviation was also added.

Page 13, Table1

Comment 5:

It will be better to mention the groups as "patients with appropriate anti-enterococcal treatment" and "patients without appropriate anti-enterococcal treatment" rather than "Treated patients" and "Untreated patients".

Response 5: Thank you for pointing this out. We have corrected it.

Page 2, lines 35–36：

 In total, 103 patients were included: 61 received treatment covering enterococci, and 42 did not.

Page 7, lines 123–126：

“Treated patients” was defined as initiating treatment in patients with antimicrobial agents that exhibit susceptibility toward the enterococci detected in ascites fluid within 48 h of acquiring ascites.

Page 12, lines 203–204：

Of these, 61 patients received treatment covering enterococci, whereas 42 did not.

Page 13, lines 230–231：

Table 1. Comparison of the clinical characteristics of the patients with and without treatment covering enterococci

Figure1

Comment 6:

As per Figure 1, if the same Enterococcus species was detected on the same day, it was counted as one case. Does it mean that if same Enterococcus species was detected in same patient on subsequent days, that patient was considered as more than one case? If latter is the true, authors need to relook into the entire statistical calculation. Please clarify.

Response 6: Thank you for your valuable comments.

If the same enterococci species was detected in the same patient the next day, it was counted as one case. No such case occurred because patients with a history of antimicrobial therapy were excluded. I indicate the exclusion criteria in the text.

Pages 6–7, lines 107–109

The exclusion criteria were age <18 years, intra-abdominal infection without microbiological culture, no intra-abdominal infection (e.g., routine drain culture after surgery), patient on antimicrobials at the time of culture collection,

Comment 7:

As per Materials and methods, patients receiving levofloxacin and meropenem were excluded from the study. However, the figure 1 mentions only meropenem treatment as one of the criteria for exclusion.

Response 7: Thank you for pointing this out. Since no patient received levofloxacin, only meropenem was listed in Figure 1. We would like to add to the text.

Page 7, line 117：No patients in this study received levofloxacin.

Our point-by-point responses to the comments and suggestions by Reviewer＃1 are listed below.

Dear Reviewer:

Thank you for your constructive comments. We have carefully revised our manuscript as suggested, and your insightful comments have helped us significantly improve the quality of our manuscript. The page and line numbers in the responses refer to the relevant parts of the main manuscript (changes marked in red font) where the text has been revised.

Reviewer #1:

Comment 1:

Authors have considered a good topic for discussion. However, a sufficient amount of scientific evidence is available on the given research question including a recent systematic review and meta-analysis by Zhang et al. (as mentioned below) with similar conclusions as that of the present study. Authors are requested to review it. Zhang J, Yu WQ, Chen W, Wei T, Wang CW, Zhang JY, Zhang Y, Liang TB. Systematic Review and Meta-Analysis of the Efficacy of Appropriate Empiric Anti-Enterococcal Therapy for Intra-Abdominal Infection. Surg Infect (Larchmt). 2021 Mar;22(2):131-143. doi: 10.1089/sur.2020.001. Epub 2020 Jun 1. PMID: 32471332.

Response 1: Thank you for your comments. Our comments on this literature can be found on pages 20–21, lines 285–298. I have made additional statements.

Page 20–21, lines 285–293：

A meta-analysis including 23 randomized controlled trials and 13 observational studies [17] showed no improvement in mortality with empiric use of anti-enterococcal agents. Most of the studies included in the meta-analysis were conducted in patients with non-severe community-onset intra-abdominal infections. Although there was no difference in mortality with empiric therapy for patients with cancer, malignancies were associated with a higher risk of enterococcal infections. Therefore, based on the results of these studies, we propose empiric administration of anti-enterococcal agents only for intra-abdominal infections in severely ill patients with cancer.

Comment 2:

The objective of the study as mentioned by the authors in the abstract is “to evaluate the impact of empiric administration of anti-enterococcal agents on the treatment of intra-abdominal infections…” However, as per the study methodology it is evident that, the study measures the impact of definitive therapy guided by culture and sensitivity report showing growth of Enterococcus species and not the empiric therapy. Hence, authors are requested to clarify it and modify the write up at relevant places in the document.

Response 2: Thank you for your very constructive remarks. By "effective empiric therapy for enterococci," I meant "treatment including enterococci as a target was initiated before the ascitic fluid culture was identified, and the patient has been treated with that antimicrobial for at least 4 days." We have added clarification where appropriate.

Page 4, lines 67–69：

Consequently, there are no definitive guidelines regarding which patients should receive empiric therapy with antimicrobial agents targeting enterococci before culture results are known.

Page 5, lines 82–85：

Therefore, this study aimed to determine whether the empirical administration of agents effective against enterococci before culture results are known is crucial for the treatment of intra-abdominal infections in immunocompromised patients with cancer.

Page 7, lines 121–125：

"The initiation of therapy that includes enterococci as a target before ascites cultures are known and continues for at least 4 days " was defined as "effective empiric therapy for enterococci." Therefore, we first divided the patients into two groups: those who received treatment covering enterococci and those who did not.

Page 7-8, lines 126–128：

Patients who did not receive treatment covering enterococci were defined as those who did not meet the above definition.

Page 8, lines 129–130：

if one type of enterococci was present and the treatment did not cover that enterococci, the patient was classified in the "untreated patients" group.

Comment 3:

It is not clear, whether the patients in untreated group received no treatment at all or were given treatment which was ‘inappropriate’ or were given treatment for less than four days.

Response 3: Thank you for your valuable comments. We excluded patients who did not receive antimicrobial therapy for intra-abdominal infections, so this study only included cases in which some type of antimicrobial agent was administered. "Untreated" includes 37 patients who did not receive antimicrobials covering enterococci and 5 patients who received antimicrobials covering enterococci for less than 4 days. We have added this information to the text.

Page 6–7, lines 107–110

The exclusion criteria were age <18 years, intra-abdominal infection without microbiological culture, no intra-abdominal infection (e.g., routine drain culture after surgery), patient on antimicrobials at the time of culture collection, absence of aggressive treatment for intra-abdominal infection, and unknown outcomes.

Page 12, lines 204–206：

Of the 42 patients, 37 had no antimicrobials covering enterococci, and 5 had antimicrobials covering enterococci for less than 4 days.

Comment 4:

Table 2 and 3 - Cancer stage – range parameter not visible.

Response 4: Thank you for pointing this out. We have corrected this.

Page 16：Table2

Page 18：Table3

Comment 5:

Kindly give full form for DHL agar plate.DHL

Response 5: Thank you for pointing this out. We have added a note.

Page 10, lines 165–166：

Depending on the results of Gram staining, a DHL (Deoxycholate Hydrogen sulfide Lactose) agar plate

Comment 6:

Since the data collection period is older, interpretative criteria used for antimicrobial susceptibility testing is quite older. A lot of revision has happened in recent years. Authors should take that into account.

Response 6: Thank you for your comment. The enterococcal ampicillin and vancomycin breakpoints have not changed from CLSI M100-S22 to the current CLSI M100 Ed33. Therefore, although this data is old, it was considered useful.

Page 10, lines 176–177：

M100-S22(2012) has not changed breakpoints with ampicillin and vancomycin by the current M100-Ed33(2023).

Comment 7:

A subgroup analysis could have been done for patients treated with appropriate antimicrobial therapy and having PS-2-4 to substantiate the claim.

Response 7: Thank you very much for your very valuable feedback. We have considered it. Due to the small number of cases in this data set, there was no difference. The p-values of Fisher's exact test for PS and mortality and PS and composite outcome for "treated patients" were both 0.64. I added to the text.

Page 21, lines 295–296：

This study did not perform a subgroup analysis of PS values and outcomes due to the small sample size.

Our point-by-point responses to the comments and suggestions by Reviewer＃2 are listed below.

Dear Reviewer:

Thank you for your constructive comments. We have carefully revised our manuscript as suggested, and your insightful comments have helped us significantly improve the quality of our manuscript. The page and line numbers in the responses refer to the relevant parts of the main manuscript (changes marked in red font) where the text has been revised.

Reviewer #2:

Comment 1:

Abstract – Line 46-47 : Retrospective analysis shouldn’t provide recommendations to change practice. It should provide recommendation to test the outcome in a prospective randomised trial to confirm the results.

Response 1: Thank you for pointing this out. We have corrected it.

Page 3, lines 45–48

The results of this study suggest that routine administration of anti-enterococcal agents for intra-abdominal infections may not be essential for all patients with cancer. To substantiate these findings, validation by a prospective randomized trial is warranted.

Page 23, lines 343–344

Future prospective randomized trials are warranted to confirm these results.

Comment 2:

In exclusion criteria , Line 109-114：Exclusion due to treatment with levofloxacin /meropenem doesn’t feel justified given their marginal/negligible activity against Enterococci. These patients might be receiving those drugs to target other organism concurrently awaiting culture reports. Even after culture reports, if they still didn’t receive appropriate treatment, they can simply be put in “no appropriate treatment” arm.

Response 2: Thank you very much for your valuable comments. In this study, only 3 patients were excluded due to levofloxacin/meropenem use. All were using meropenem, and there were no levofloxacin cases. So, since the number of patients excluded was very small, the impact on the results is considered to be minimal. In addition, in this research method, cases were enrolled if they were changed to an antibacterial agent other than meropenem within 48 hours of culture collection and were treated with the changed antimicrobial agent for at least 4 days. In fact, there were no such cases. We added an addendum to the text.

Page 23, lines 330–333

Also, our study excluded meropenem and levofloxacin, which have negligible activity against enterococci. However, only three cases were excluded, and it is unlikely they could have influenced the study results.

Comment 3:

Table 2 –Univariate and multivariate analysis – the factor “multiple micro-organisms” should also be tested on univariate analysis. This might be relevant as 1) it might indicate more severe sepsis 2). It was unequally distributed in both arms.

Response 3: Thank you for pointing this out. We were unable to analyze the data because all deaths fell into the "multiple microorganism" category. It has already been mentioned in the text.

Page 17, lines 251–253

It was impossible to analyze the multiple microorganisms isolated because all patients who reached the primary outcome had multiple microorganisms detected in their ascitic fluid.

Comment 4:

Rather than taking Pitt bacteremia score cut-off of 2, different cut-offs can be tested in univariate analysis to see if value above a certain cut-off is associated with increased mortality.

Response 4: Thank you for your valuable feedback. Splitting the Pitt bacteremia score between 0 and 1 or higher did not change the results of the univariate analysis. I have included the results below.

＜univariate analysis＞

Risk factors for mortality (Please see the table in the docx file in Response to Reviewers_2.)

Risk factors for composite outcomes (Please see the table in the docx file in Response to Reviewers_2.)

Comment 5:

Line 242-246 : The word “ outcome” isn’t clear. What exactly the authors mean by the term “outcome”.

Response 5: Thank you for your guidance. The outcome here means mortality in the primary outcome.

Page 17, lines 250–253:

Enterococcal bacteremia could not be analyzed because none of the patients who achieved the primary outcome had enterococcal bacteremia. It was impossible to analyze the multiple microorganisms isolated because all patients who reached the primary outcome had multiple microorganisms detected in their ascitic fluid.

Comment 6:

As there were only 3 patients with hematologic malignancies (which represents a totally different profile of immunocompromised patients compared to solid cancers), to make the study more homogenous and generalizable , it would be better to include only solid cancer patients. As such 3 patients is too small a sample size for representing hematological malignancies in the study.

Response 6: Thank you for your very valuable advice. As you pointed out, there were only 3 patients with hematologic malignancies, so it is difficult to generalize to all cancers. However, since the overall sample size itself was small even if hematologic malignancies were included, we included hematologic malignancies to maintain statistical quality. I added this to the Limitation section.

Page 22, lines 324–328:

Thus, there were limitations in assessing differences in outcomes between solid tumors and hematologic malignancies, as well as differences in outcomes in various cancer types. Concerning patients with hematologic malignancies, for which the sample size was small, the results of this study are not well grounded to be generalized as is.

Comment 7:

Conclusion : Its not advisable to recommend a practice change just based on one retrospective study which doesn’t rule out any inherent bias due to retrospective nature of study design, unless a good quality systematic review of all pooled retrospective data also draw same conclusion. Here, the conclusion should suggest a prospective randomised trial to confirm the observed findings.

Response 7: Thank you for your instruction. I have added it to the text.

Page 23, lines 343–344

Future prospective randomized trials are warranted to confirm these results.

---

## [Decision Letter · Decision Letter 1]

26 Dec 2023

PONE-D-23-31414R1To treat or not to treat: Assessing the role of anti-enterococcal therapy for intra-abdominal infections in patients with cancerPLOS ONE

Dear Dr. Itoh,

Thank you for submitting your manuscript to PLOS ONE. After careful consideration, we feel that it has merit but does not fully meet PLOS ONE’s publication criteria as it currently stands. Therefore, we invite you to submit a revised version of the manuscript that addresses the points raised during the review process.

We look forward to receiving your revised manuscript.

Kind regards,

Arghya Das, MD

Academic Editor

PLOS ONE

Journal Requirements:

Reviewers' comments:

Reviewer's Responses to Questions

**Comments to the Author**

1. If the authors have adequately addressed your comments raised in a previous round of review and you feel that this manuscript is now acceptable for publication, you may indicate that here to bypass the “Comments to the Author” section, enter your conflict of interest statement in the “Confidential to Editor” section, and submit your "Accept" recommendation.

Reviewer #1: All comments have been addressed

Reviewer #2: All comments have been addressed

2. Is the manuscript technically sound, and do the data support the conclusions?

Reviewer #1: Partly

Reviewer #2: Yes

3. Has the statistical analysis been performed appropriately and rigorously? 

Reviewer #1: I Don't Know

Reviewer #2: Yes

4. Have the authors made all data underlying the findings in their manuscript fully available?

Reviewer #1: Yes

Reviewer #2: Yes

5. Is the manuscript presented in an intelligible fashion and written in standard English?

Reviewer #1: No

Reviewer #2: Yes

6. Review Comments to the Author

Reviewer #1: Thanks for considering revision of the manuscript after the comments. However, the point of empiric versus definitive therapy is still not addressed very convincingly.

Reviewer #2: (No Response)

7. PLOS authors have the option to publish the peer review history of their article (what does this mean?). If published, this will include your full peer review and any attached files.

Reviewer #1: No

Reviewer #2: **Yes: **Babita Kataria

---

## [Author Response · Author response to Decision Letter 1]

31 Dec 2023

RESPONSE TO THE REVIEWER’S COMMENT

Our response to the comment by Reviewer＃1 is provided below.

Dear Reviewer:

Thank you for your constructive comments. We have carefully revised our manuscript as suggested, and your insightful comment has helped us significantly improve the quality of our manuscript. The page and line numbers in the below response refer to the relevant parts of the main manuscript where the text has been revised (the changes are marked in red font).

Reviewer #1:

Comment 1:

Thanks for considering revision of the manuscript after the comments. However, the point of empiric versus definitive therapy is still not addressed very convincingly.

Response 1: Thank you for highlighting this issue. We have changed the expression "empiric therapy" to "initial antimicrobial therapy" and made the corresponding changes to the relevant parts of the manuscript as indicated below. In addition, we have included an additional section of text providing further information, which is also provided below.

Page 2, lines 27–30:

In this study, we evaluated the impact of the initial antimicrobial therapy administration of anti-enterococcal agents on the treatment of intra-abdominal infections in patients with cancer in whom enterococci were isolated from ascitic fluid cultures.

Page 3, lines 46–48:

The results of this study suggest that the initial routine administration of anti-enterococcal agents for intra-abdominal infections may not be essential for all patients with cancer.

Page 5, lines 83–86:

Therefore, this study aimed to determine whether the initial administration of agents effective against enterococci before culture results are known is crucial for the treatment of intra-abdominal infections in immunocompromised patients with cancer.

Pages 7–8, lines 121–130:

Patients were defined as "receiving effective initial therapy for enterococci" if antimicrobial therapy was initiated within 48 hours of ascitic fluid collection, the enterococci detected in the ascitic fluid were susceptible to that antimicrobial therapy, and treatment was continued for at least 4 days. Patients were divided into two groups, with the group that received the above-defined effective initial therapy against enterococci classified as "treated patients" and the group that did not receive treatment defined as "untreated patients”. In cases where multiple types of enterococci were detected in the ascitic fluid, if one or more types of enterococci were present that were not covered by appropriate treatment, the patient was classified as part of the "untreated patients" group.

Pages 20–21, lines 291–293:

Therefore, based on the results of these studies, we propose the empiric administration of anti-enterococcal agents from the beginning only for intra-abdominal infections in severely ill patients with cancer.

Page 23, lines 341–343:

Our findings suggest that not all cancer patients require initial antimicrobial therapy against enterococci and that treatment should be considered for severe cases and patients with PS2 or higher.

---

## [Editor Report · Decision Letter 2]

17 Jan 2024

To treat or not to treat: Assessing the role of anti-enterococcal therapy for intra-abdominal infections in patients with cancer

PONE-D-23-31414R2

Dear Dr. Itoh,

We’re pleased to inform you that your manuscript has been judged scientifically suitable for publication and will be formally accepted for publication once it meets all outstanding technical requirements.

Kind regards,

Arghya Das, MD

Academic Editor

PLOS ONE

Additional Editor Comments (optional):

Few very minor modification should be done in the copy editing and authors' proofing stages.

Table 1: Add "years" to "Age" as specific unit in Table 1.

Introduction: Replace the term "nosocomial" with more recent terms such as "hospital-acquired" or, "healthcare-associated"

Results:

"Enterococcal bacteremia could not be analyzed because none of the patients who achieved the primary outcome had enterococcal bacteremia"

Rephrase the above sentence as "Enterococcal bacteremia could not be analyzed because none of the patients who had the primary outcome suffered from enterococcal bacteremia"

"Enterococcal bacteremia could not be analyzed because none of the patients who achieved composite outcomes had enterococcal bacteremia"

Rephrase the above sentence as "Enterococcal bacteremia could not be analyzed because none of the patients who had composite outcomes suffered from enterococcal bacteremia"
---

## [Editor Report · Acceptance letter]

30 Jan 2024

PONE-D-23-31414R2 

PLOS ONE

Dear Dr. Itoh, 

I'm pleased to inform you that your manuscript has been deemed suitable for publication in PLOS ONE. Congratulations! Your manuscript is now being handed over to our production team.

Kind regards, 

on behalf of

Dr. Arghya Das 

Academic Editor

PLOS ONE